# Natural Coatings and Surface Modifications on Magnesium Alloys for Biomedical Applications

**DOI:** 10.3390/polym14235297

**Published:** 2022-12-04

**Authors:** Diego Cuartas-Marulanda, Laura Forero Cardozo, Adriana Restrepo-Osorio, Patricia Fernández-Morales

**Affiliations:** 1Grupo de Investigación sobre Nuevos Materiales (GINUMA), Semillero de Investigación en Textiles (SI Textil), Escuela de Ingenierías, Universidad Pontificia Bolivariana, Circular 1 # 70-01, Medellín 050031, Colombia; 2Semillero de Investigación en Nuevos Materiales (SINUMA), Escuela de Ingenierías, Universidad Pontificia Bolivariana, Circular 1 # 70-01, Medellín 050031, Colombia; 3Grupo de Investigación sobre Nuevos Materiales (GINUMA), Semillero de Investigación en Nuevos Materiales (SINUMA), Escuela de Ingenierías, Universidad Pontificia Bolivariana, Circular 1 # 70-01, Medellín 050031, Colombia

**Keywords:** magnesium alloys, degradation, natural polymers, surface modification, osseointegration, bioresorbable

## Abstract

Magnesium (Mg) alloys have great potential in biomedical applications due to their incomparable properties regarding other metals, such as stainless steels, Co–Cr alloys, and titanium (Ti) alloys. However, when Mg engages with body fluids, its degradation rate increases, inhibiting the complete healing of bone tissue. For this reason, it has been necessary to implement protective coatings to control the rate of degradation. This review focuses on natural biopolymer coatings used on Mg alloys for resorbable biomedical applications, as well as some modification techniques implemented before applying natural polymer coatings to improve their performance. Issues such as improving the corrosion resistance, cell adhesion, proliferation, and biodegradability of natural biopolymers are discussed through their basic comparison with inorganic-type coatings. Emphasis is placed on the expected biological behavior of each natural polymer described, to provide basic information as a reference on this topic.

## 1. Introduction

Implant devices are used to treat a fracture or osteoarthritis, and for other orthopedic fixations (spinal, knee, hip) [1]. In recent years, biodegradable alloy materials such as Fe, Zn, and Mg have received considerable attention due to their comparative advantages in the field of orthopedic implants concerning metals that are typically used (stainless steel, cobalt–chromium (Co–Cr) alloys, and Ti alloys) [2]. Biodegradable Mg-based alloys used as orthopedic implants stand out for their low density, their similar mechanical properties to native bone, and their excellent biocompatibility. Mg is present in several enzymatic reactions; it influences heart, neurological, and digestive functions, and it also promotes the growth of human bones. Under this perspective, Mg is considered an essential element for the biological functions of the human body and with great potential for its use in implants for osseointegration.

However, one of the major limitations of Mg alloys is their high corrosion rate compared to those reported for Zn and Fe [3]. Currently, most studies are focused on the control of its rapid degradation, where the use of ceramic coatings, polymeric coatings, or the combination of both (hybrid coatings) has been addressed. In the search for a coating not only as a layer to control implant degradation but also to contribute to the rapid generation of bone structures, the three most relevant concepts to consider are osteoinduction, osteoconduction, and osseointegration, according to its relationship with bone morphogenic proteins (BMP), bone growth factors, and direct bone anchorage, respectively. Conceptually, these three terms are defined as follows: osteoinduction refers to the process by which osteogenesis is induced, osteoconduction means that the bone grows on a surface, and osseointegration is defined as the contact interface between the living bone and implant, which is the direct anchorage of an implant by the formation of bony tissue around the implant [4].

It can be said that the success of a Mg-based implant is subject to the performance of the coating. In the case of Mg alloy implants for osseointegration, the coatings should be not only biocompatible but also biodegradable, osteoconductive, osteoinductive, and, if possible, with a porous structure similar to bones. Among the strategies to control its rapid degradation, most studies are focused on the use of ceramic coatings, polymeric coatings, or a combination of both (hybrid coatings). Generally, both inorganic coatings such as calcium phosphates (hydroxyapatite is the most common) and organic coatings (natural polymers) are biocompatible and biodegradable, have very good osseointegration, promote cell and bone growth on the surface of the implant, as well as repair and tissue growth. Ceramic coatings also offer chemical and mechanical properties close to the bone, with high solubility and bioresorbability, but, at the same time, they have poor mechanical properties (low tensile strength, low fracture toughness, and poor impact resistance) and brittle layers as their main drawbacks [5,6]. Brittle layers of ceramic could generate complications that lead to implant rejection due to poor osseointegration [7]. Deficiency at the bone–implant interface leads to aseptic loosening and can also generate implant debris, such as wear particles or coating detachments. In general, debris particles inhibit direct prosthesis–bone contact; cannot be easily phagocytosed, causing an inflammatory response; and are more susceptible to microbial colonization [8,9]. In this sense, an implanted device must retain its mechanical integrity until the bone tissue is recovered, and then gradually dissolve to be replaced by the osteogenic tissue [10].

On the other hand, natural polymer coatings have a unique combination of characteristics, such as high cytocompatibility, biodegradability, and the ability to trigger a reaction that is as physiologically similar as possible to the surrounding tissues, leading to the regeneration of new bone. Either using only natural polymer layers or combined with ceramic base layers as hybrid coatings, one would expect satisfactory performance in functions such as osteoinduction and osteoproliferation (osteoblasts). This means that facilitating and promoting cell proliferation can be expected to result in a higher regeneration rate, facilitating the biological fixation of the implant in the human body. Moreover, natural polymers provide the possibility of the delivery of drugs, biomolecules, proteins, or stem cells. All these aspects place natural polymers in the spotlight as a very attractive alternative as a coating for Mg alloy implants [6]. Mg and its alloys have received great attention due to their unique biological properties for biomedical applications as implants [11]. Mg implants have high bioactivity, excellent biocompatibility, and biodegradability and minimize the risk associated with a second surgery and the linked costs of medical care and trauma to the patient [12]. Approximately 20 g of Mg is always present in the average 70 kg human body [13]; Mg is naturally present in bone tissue and is involved in many metabolic reactions and biological mechanisms [14,15]. The Young’s modulus and the compressive yield strength of Mg are similar to those of natural bone, making magnesium and its alloys notable candidates for temporary orthopedic applications [12]. The density and mechanical properties of Mg compared to natural bone and typical metals used for implants are shown in Table 1.

The degradation of Mg is an advantage that makes it one of the best candidates for use in bioabsorbable implants for osseointegration. This phenomenon occurs at high speed, while hydrogen release could compromise bone fracture healing and lead to surrounding soft tissue inflammation. However, if the Mg is exposed to typical atmospheric conditions, it develops a gray oxide film of magnesium hydroxide (Mg(OH)_2_), which slows the corrosion phenomenon. The Mg(OH)_2_ layer is slightly soluble in water, meaning that severe corrosion occurs in aqueous physiological environments because the Mg (OH)_2_ reacts with the chloride ions present in the human body to form magnesium chloride and hydrogen gas [16]. Various factors have an influence on Mg’s corrosion in body fluids, such as pH, electrolyte concentrations and types of ions, protein adsorption on the orthopedic implant, and the biochemical activities of surrounding tissues; see Figure 1 [19]. The chemical reactions generated during the corrosion of Mg are shown as follows [16]:(1)$$\mathrm{Mg}\;\left(s\right)+2H_{2}O\to \mathrm{Mg}\;{\left(\mathrm{OH}\right)}_{2}\;\left(s\right)+H_{2}\;\left(g\right)$$
(2)$$\mathrm{Mg}\;\left(s\right)+2{\mathrm{Cl}}^{-}\left(\mathrm{aq}\right)\to {\mathrm{MgCl}}_{2}$$
(3)$$\mathrm{Mg}\;{\left(\mathrm{OH}\right)}_{2}\;\left(s\right)+2{\mathrm{Cl}}^{-}\to {\mathrm{MgCl}}_{2}$$

The rapid corrosion of Mg implants and its alloys under physiological environments before bone formation and non-uniform degradation, combined with hydrogen release, make inevitable the use of conventional devices [23]. Studies have been conducted to tackle the problem of the corrosion of Mg implants. One of the most common approaches has been the addition of alloying elements to enhance the inherent performance of pure materials. However, the complicated composition will lead to high production costs. In addition, some elements, such as beryllium and nickel, should be strictly controlled and even avoided as they may potentially cause adverse effects on the human body [24]. Heat treatment strategies and their influence on the corrosion properties of Mg–Zn–Ga alloys have been studied to investigate their usefulness for osteosynthesis applications, with promising results, such as a low corrosion rate and good mechanical properties [25]. Other techniques, such as surface modification and coatings with ceramic, polymeric, and hybrid materials, are effective in controlling the degradation rate of Mg [26]. The effects of these treatments on the degradation resistance of Mg or its alloys can be measured in different approaches. Potentiostatic and potentiodynamic polarization curves, transient currents, and electrochemical impedance spectroscopy are the most common techniques to study corrosion problems in Mg alloys [27]. Generally, the mass loss of substrates and evolution volume of hydrogen are two frequently used methods for in vitro measurements of degradation rates, and electrochemical parameters may also be used to predict their degradation properties [24].

To maintain the ideal degradation behavior of Mg, surface coatings have become one of the preferred strategies to provide acceptable control. Coatings can contribute to retarding corrosion and isolating the material from corrosive fluids [10]. Through modification of the implant surface by varying topography and coatings, it is possible to enhance the osseointegration [7,28]. The surface features of implants, such as wettability, porosity, and roughness, affect the biological and cellular response, improving cellular proliferation and differentiation [7]. A coating must satisfy the following criteria: it must not trigger an immune response, it must be osteoconductive and osteoinductive, it must have adequate mechanical stability and antimicrobial properties, and it must be able to recruit stem cells [29].

As mentioned above, polymeric coatings are of high interest to enhance the corrosion resistance, abrasion, and wear properties of Mg alloy. These types of coatings can also lead to cellular responses such as adhesion and proliferation and show high degradation capacity along with biocompatibility, self-healing, and drug delivery [30]. There are two large groups of polymeric coatings: synthetic and natural polymers. Natural polymers present advantages such as their excellent biocompatibility due to their biomimetic nature, as well as their biological activity [10,19,31].

It is interesting for this review to highlight the potential offered by natural polymers for use as a coating for Mg-based bioabsorbable implants. The biocompatible natural polymers have a series of additional highly attractive characteristics required for better implant performance, such as attracting cell adhesion, stimulating a cell response, better bio-performance, and improved corrosion resistance [7]. Moreover, they provide a biochemical environment similar to natural bone, in some cases having the possibility to release growth factors or drugs [32,33,34]. These materials present a low risk of immune response, non-toxicity, hydrophilicity, and few side effects. Their degradation products are amino acids, which possess extracellular components, appropriate porosity, and good pore interconnectivity for blood vessel growth [35,36]. Furthermore, they enable biodegradation at the desired rate and a limited barrier function [13].

Mg and Mg alloys such as AZ31, AZ91, AM50, ZK60, or WE43 are suitable materials for biomedical applications, and when they are coated with natural polymers, they can biodegrade, while also providing self-healing drug delivery and osteoinduction [7,14,34,37]. These properties make them suitable for applications in which the temporary presence of an implant is necessary [13]. In particular, natural polymer coatings on Mg and Mg alloys have been studied in orthopedic implants, cardiovascular stents, antibacterial surfaces, drug delivery, and so on [38].

The application of this type of coating is interesting considering that the ability of bone to regenerate is due to the presence of stem cells, which can differentiate into fibroblastic, adipogenic, reticular, and osteogenic cells [8]. The functionality and differentiation of osteoblast cells depend on the morphological and mechanical features of the extracellular matrix (ECM), whose structure is composed of proteins such as collagen (Coll) and elastin. A surface with similar chemical and morphological properties will provide scaffold support as a consequence of tissue formation [34]. These mentioned conditions are feasible to replicate using either natural polymer coatings on hydroxyapatite layers (hybrid coatings) or only the functionalized polymer layers. In either case, the key material remains the natural polymer.

For example, Liangjian C. et al. developed a chitosan (CS) coating on a Mg composite for implant applications. They evaluated the in vitro degradation behavior in simulated body fluid (SBF) and in vitro cytotoxicity in L-929 cells. The coated sample showed lower hydrogen release than the uncoated one, and the CS coating improved the cytocompatibility of the magnesium composite [39]. On the other hand, Elkamel S. et al. compared different nanocoatings on Mg staples. Coatings with gelatin (GEL), CS, and cellulose acetate phthalate (CAP) nanoparticles with the non-steroid anti-inflammatory drug (NSAID) diclofenac were prepared. The evaluation of corrosion and hydrogen release tests showed that the GEL nanoparticle coating had higher corrosion resistance for 7 days. Nevertheless, CAP nanoparticles represented the most protective coating at a longer immersion time of 20 days. Diclofenac drug release was found to suppress inflammation and help to heal tissue [40].

Minting D. et al. fabricated a vancomycin (Van)-loaded sodium alginate hydrogel coating on a micro-arc oxidation (MAO)-treated Mg alloy. The tests indicated that the corrosion resistance of the coated Mg alloy was improved, and the sample presented excellent antibacterial properties and hemocompatibility as well [41]. Zhao N. et al. prepared a Coll and hydrofluoric acid (HF) coating on two rare-earth-based Mg alloys (MgYZrRE and MgZnYZrRE) for cardiovascular stent applications. Results indicated that the coating enhanced biodegradation and cytocompatibility [42].

It has been found that natural polymeric coatings, together with surface modification, improve the surface properties of Mg alloys, providing a protective barrier to retard corrosion degradation [43]. In this review, some modification techniques implemented before the use of natural biopolymer coatings are discussed, as well as natural polymer coatings used on Mg alloys for biomedical applications. These developments have great potential in improving the corrosion resistance, cell adhesion, proliferation, and biodegradability of Mg-based implants.

## 2. Surface Modification

Mg implants must satisfy two fundamental characteristics to guarantee tissue healing: biocompatibility and resistance to degradation. Biocompatibility is a characteristic in which the material can be metabolized in the human body. An alternative to guarantee resistance to degradation is surface modification. It can be classified into three categories: chemical modification, physical modification, and a combination of these, which seeks the elimination of the shortcomings and supplementation of the advantages of the individual techniques; see Figure 2 [24].

### 2.1. Chemical Modification

Conversion coatings are in situ chemical or electrochemical interactions of the metal scaffolds with the environment, which leads to the formation of a superficial layer of substrate metal oxides, chromates, phosphates, or other compounds that are chemically bonded to the surface [44,45]. These types of coatings are among the most cost-effective methods; therefore, they are widely used to provide a barrier between the metal and its environment [46].

Conversion coatings include chromates, phosphate/permanganate, and fluorides [45]. The content of chromates has been reported to cause environmental problems and human body toxicity [47]. Other techniques include alkaline treatment, acid etching, electron beam treatment, and micro-arc oxidation coating [24].

#### 2.1.1. Fluoride Treatment

The fluoride chemical conversion process is considered a simple and low-cost technique to obtain protective coatings in Mg alloys [48]. Mg reacts with hydrofluoric acid (HF) to form MgF_2_ via a displacement reaction, forming a barrier coating on Mg with increased resistance to polarization [49]. Fluoride is an essential natural component in the human diet; it is required for normal dental and skeletal growth. Moreover, it is one of the few known agents that can stimulate osteoblast proliferation and increase new mineral deposition in cancellous bones [48].

Several authors have studied biomedical coatings containing fluoride on Mg alloys. It has been observed that the coating process is usually performed by the immersion of Mg or Mg alloy substrates in 40% or 48% HF [13]. A chemical conversion treatment in HF was applied on the surface of an AZ31B Mg alloy to reduce its degradation rate. The biodegradation kinetics of the AZ31 alloy evaluated by EIS confirmed that the fluoride coating offered effective protection that delayed the degradation process [50,51]. Another important result was the corrosion products formed in the coated materials, which were compounds rich in calcium and phosphorus, necessary for bone health [50]. Furthermore, the bonding strength between the fluoride coating and AZ31B substrate was over 43.2 MPa, while the minimum bonding strength of 22 MPa is required for a coating on medical implants according to ASTM 1147-F [51].

#### 2.1.2. Alkaline Treatment

Alkaline treatment consists of a passive layer coated on the Mg sample by using a NaOH solution [52]. It was reported that by immersing the implant surface in 1 M NaOH for 24 h, the reactivity of the surface was reduced due to the formation of a thin passive layer of Mg (OH)_2_ [53]. The study also investigated the corrosion behavior of heat- and alkali-treated magnesium samples in simulated body fluid (SBF). It showed that the mass of the samples remained almost constant during the 14-day tests, which implies that this treatment has good corrosion resistance. Moreover, the pH values increased slowly compared to those of the untreated sample [54].

### 2.2. Physical Modification

Physical modification techniques are less often used on metals such as Mg. They do not form chemical bonds between the substrate and the outer surface. The physical coatings are used to modify the structure by introducing physical processing techniques or new phases such as inorganic compounds and polymers. There are different techniques, such as laser modification, metal oxide implantation, organic coatings, and apatite coatings [24].

One of the latest techniques used for surface modification is called directed plasma nano-synthesis (DPNS) and it seeks to provide a bioactive and bioresorbable interface for Mg foams. A study in which DPNS was used in open-porous Mg foams to modify the surface chemistries and topographies was published. It obtained a surface modification strategy that adjusts the interaction of the material and the environment without using a coating that affects the geometry and properties of the porous material [55].

A novel technique used before a natural polymer coating application is surface activation by ultraviolet ozone (VUV/O_3_). The fibroin-coated MgZnCa alloy prepared by O_2_ and VUV/O_3_ plasma activation showed improved storage properties and in vitro corrosion resistance compared to the bare MgZnCa alloy. Furthermore, it was confirmed that the structures prepared via the activation of VUV/O_3_ possessed enhanced biocompatibility, bioactivity, and biosafety in systematic cell adhesion and cytotoxicity experiments [56].

## 3. Natural Polymeric Coating

The use of coatings is effective in decelerating degradation and reducing hydrogen evolution [23]. Natural polymers are classified into proteins (Coll, GEL, albumin, and silk fibroin), polysaccharides (CS, alginate, and cellulose), and weak organic acids. These polymers have been studied as coatings on the surface of Mg and its alloys because they provide barriers to limit direct contact between the surface of Mg and the aqueous biological environment [12]. Natural polymers usually contain biofunctional molecules that guarantee bioactivity [57] and non-immunogenic properties [58]. Another advantage of natural coatings is the ability to add functional properties to the implant surface, adapt the composition and structure of the coating [12], serve as templates for cell attachment and growth, and foment cellular responses; they have been utilized in biomedical fields (e.g., soft tissue engineering) for a long period [10]. Modified polymeric coatings can act as host reservoirs for corrosion inhibitors and can provide more effective active corrosion protection with self-healing properties [59].

### 3.1. Polysaccharides

#### 3.1.1. Chitosan

Chitosan (CS) is a positively charged natural polysaccharide composed of N-acetyl-D-glucosamine linked to β-(1→4) that is vulnerable to biodegradation [60]. CS is a derivative of chitin, the major component of the exoskeletons of crustaceans and the cell walls of fungi [61]. It is the second most abundant natural biopolymer after cellulose and is widely distributed both in the animal and plant kingdom [58]. CS is characterized by its cytocompatibility, biodegradability, and non-toxicity [57]. It can serve an important function as an adhesive basal matrix for growing cells during the peri-implant healing process and enhances the corrosion resistance of biodegradable metals [19]. It is a biocompatible material that breaks down slowly into harmless products (amino sugars) that are absorbed by the body [58]. Depending on the source and preparation procedure, its molecular weight may range from 300 to over 1000 kD, with a degree of deacetylation from 30% to 95%. CS is normally insoluble in aqueous solutions above pH 7. However, in dilute acids (pH 6.0), the protonated free amino groups on glucosamine facilitate the solubility of the molecule [62,63].

CS has become a coating biomaterial for Mg and its alloys due to its interesting characteristics for biomedical applications: minimal foreign body reaction, intrinsic antibacterial nature, and the ability to be molded into various geometries and forms, such as porous structures. Among the aspects of greatest interest, its possible use in porous structures makes it a suitable alternative for applications in osteoconduction [64]. CS coatings exhibit good biocompatibility properties, low degradation, and the potential to swell and dehydrate depending on the composition and environment [65]. It has high resistance to compression and facilitates cell propagation and proliferation [66,67].

The molecular weight and number of layers of CS coatings have been shown to influence substrate corrosion. Adding a layer of CS to AZ91E alloy decreased its corrosion rate in an artificial sweat solution. It was found that the corrosion rate decreased when increasing the polymer concentration from 5% to 15% [68]. Likewise, it was observed in Mg-1Ca alloys that the best results were obtained when six layers of CS coating were applied with a molecular weight of 2.7 × 10^5^, because the hydrogen evolution rate was the lowest [69]. In vivo studies revealed that a chitosan coating on Mg and its alloys is favorable for decreasing the in vivo degradation of the substrate. Nevertheless, the interfacial adhesion between the CS coating and the substrate is weak [70].

#### 3.1.2. Alginate

Alginate is an anionic hydrophilic polysaccharide that is distributed mainly in the cell walls of bacteria and brown algae [71]. Alginate contains linked (1–4) β-D-mannuronic acid (M) and α-L-guluronic acid (G) monomer blocks that are covalently linked and arranged in various forms: consecutive G sequence, consecutive M sequence, and alternate MG sequence [72]. Algal alginates usually possess high content of G blocks and are used for biomedical applications, while alginates with bacterial sources are M-enriched, immunogenic, and show greater potency to induce cytokine production [71].

Alginate is interesting as a biomaterial because it has a similar structure to the extracellular matrix (ECM) that supports cell adhesion with chemical modifications. In addition, it is biocompatible and biodegradable, as well as facilitating the administration of drugs [73]. Multilayer coatings carrying sodium alginate (ALG), CS, and mechano-growth factor (MGF) were applied on a fluoride-pretreated ZEK100 magnesium alloy. The fatigue life of the coated ZEK100 was found to be slightly greater than that of the uncoated samples after one (1) day of immersion. Furthermore, it was observed that after immersion for 14 days, the total corrosion rates for coated and uncoated samples were approximately 6.452 g/m^2^d and 7.389 g/m^2^d, respectively [74].

#### 3.1.3. Cellulose and Derivates

Cellulose is the most abundant natural polymer in the biosphere [75]. It is a linear chain of glucose molecules and has a flat ribbon conformation [76]; such units are linked by β-1,4-glycosidic [75]. Cellulose can be obtained from a variety of sources, such as wood, seed fibers, bast fibers, marine animals, algae, fungi invertebrates, and bacteria [77]. It has excellent features that make it helpful; for example, it is a renewable homopolymer that is biodegradable, sustainable, non-toxic, and highly biocompatible [78].

A study reported the use of a PLA–cellulose nanocomposite as a coating on AZ31 magnesium alloy. Different values of nanocellulose (CNs) were evaluated: 1, 5, and 10% *w*/*w* added to PLA coating and applied by the immersion method. The presence of CNs was observed to increase the corrosion resistance of the nanocomposite, and the best result was obtained at 5% *w*/*w* of CNs. The contact angle test showed an increase from 89° to 108°, which indicated hydrophobicity behavior. Higher content of CNs (from 5% w to 10% w) reduced the corrosion resistance of the PLA coating, and the presence of CNs increased the mechanical properties of the coating [79].

Another work studied the usage of cellulose acetate (CA) coated on Mg alloy for orthopedic applications. The authors synthesized Mg-1Ca-0.2Mn-0.6Zr% *w*/*w* and coated it via the dip coating method in a solution of CA in N,N′-dimethylformamide. The potentiodynamic polarization test showed that the CA coating improved the corrosion resistance of the Mg alloy, and the results proved its good cytocompatibility concerning cell adhesion, viability, the promotion of osteogenic differentiation, and proliferation [80].

#### 3.1.4. Hyaluronic Acid

Hyaluronic acid (HA) is a carbohydrate with a poly repeating disaccharide structure [(1→3)-β-D-GlcNAc-(1→4)-β-D-GlcA-] [81]. HA forms a smaller part of the extracellular matrix (ECM) but has the significant advantage of structural conservation regardless of the source and is therefore nonallergenic [82]. It is a natural biopolymer that possesses numerous functions within the body, including wound repair, cell migration, and cell signaling. Due to its versatility, it has been investigated in a number of fields, such as tissue engineering and cancer treatment [83].

The corrosion behavior of HA and cerium multi-layer films on an Mg implant was evaluated. Ce(NO_3_) was used to form an electrodeposition layer, and HA was coated through hydrothermal treatment. It was found that the polymer coating not only contributed to the initial corrosion resistance but also helped to confer corrosion resistance during local damage and biodegradation. The cell viability of osteoblasts did not show toxicity, and the cells presented the highest differentiation when treated with the HA [84].

To improve the local corrosion caused by plasma electrolytic oxidation (PEO) treatment, the formation of additional layers is used. HA and carboxymethylcellulose (CMC) have been studied for this purpose. The electrochemical corrosion test showed improved corrosion resistance with the HA coating. The HA/CMC composite layer reduces osteoblast proliferation and cell stress, promoting bone formation around the implant [85].

#### 3.1.5. Chitin

Chitin (CHN) is the most abundant natural polymer found in the inner cells of arthropods’ endoskeletons. It is a copolymer of N-acetyl-glucosamine and N-glucosamine units randomly or in blocks distributed throughout the biopolymer chain [86]. CHN with hydroxyapatite was reported to be deposited on AZ91 magnesium alloy via a dipping method. The results showed that the corrosion current density at 180 and 420 min was higher than in the uncoated ones [87]. Although CHN is rarely investigated as a coating for Mg alloys, some articles report its use on other metals, such as titanium and zinc. CHN was reported as a coating to improve the corrosion resistance of Zn alloy, but, in this case, the CHN was in form of CHN/Ni-doped ZnO. The coating improved the anticorrosion performance by enhancing the surface roughness and hydrophilic nature of the Zn alloy [88,89].

#### 3.1.6. Heparin

Heparin (HEP) is a natural polysaccharide that belongs to the family of glycosaminoglycans present in mast cells [90]. It is constituted by alternating disaccharide sequences of hyaluronic acid and a hexosamine [91]. The main repeating disaccharide unit in HEP consists of (1→4)-α-D-6-O-N-sulfoglucosamine (1→4)-linked to an α-L-2-O-sulfated iduronic acid, short under sulfated domains [92], and it is well known for its role as an anticoagulant [93].

A study developed an anticorrosive silane coating in which bare AZ31 Mg alloy and Mg-B–A-HEP were compared. The surface modification could control the Mg’s corrosion and inhibit platelet adhesion [94]. In another study, silk fibroin blended with heparin and GREDVY (Gly–Arg–Glu–Asp–Val–Tyr) was also reported to improve the corrosion resistance, blood compatibility, and endothelialization. The coating showed better corrosion resistance, reduced platelet adhesion and hemolysis rate, prolonged APTT, TT, and PT time, and better biocompatibility [95]. HEP and carboxymethyl chitosan were used on alkali-treated Mg to increase the hemocompatibility and antibacterial activity. The functionalized coating exhibited superior corrosion resistance and excellent anticoagulation properties and an antibacterial effect [96].

### 3.2. Proteins

#### 3.2.1. Collagen

Coll is the most abundant protein in the human body and the main component of most connective tissues. It has a triple helix structure of three polypeptide subunits, known as α chains [97,98]. Currently, 28 types of Coll have been identified; I, II, III, and V are the main ones that contain the essential parts of collagen in bones, cartilage, tendons, skin, and muscles [99]. The different Coll types are characterized by considerable complexity and diversity in their structure, their splice variants, the presence of additional, non-helical domains, their assembly, and their function. The most abundant and widespread family of Coll, with approximately 90% of the total Coll, is represented by the fibril-forming collagens [100]. They can be extracted from different sources, such as the rat tail tendon, the Achilles tendon of bovine origin, bovine skin, pig skin, and the skin of the human corpse [101].

Coll is an attractive biomaterial due to its low immunogenicity, biocompatibility, biodegradability, and ability to form fibers with high tensile strength, which is why it has been used as a biomedical material [102]. It was reported that Mg–Zr–Ca implants were coated with collagen type-I (Coll-I) extracted from rat tails to evaluate their rate and efficiency of bone mineralization and implant stabilization, and it was observed that the Coll-I coating improved the surface energy and hydrophobicity of these alloys and strongly influenced the protein-binding capacity on the surface of the alloy, leading to better osteoblast activity. In addition, it improved the rate of osseointegration, with a good level of new bone formation after only one month of implantation [103].

#### 3.2.2. Serum Albumin

Serum albumin (SA) is a protein present in the circulatory systems of various organisms. It serves an important function in maintaining osmotic blood pressure, drug disposition, and efficacy. SA carries a wide range of nutrients, metabolites, drugs, and ions in the blood [104]. It has wide clinical and biochemical applications [105].

Human serum albumin (HSA) is multifunctional and the most abundant protein in plasma, having a life cycle of 20–25 days [106]. Bovine serum albumin (BSA), which is a homolog of has, is utilized in pharmacokinetic and affinity tests of drugs as a replacement for HSA, because it is much cheaper and much easier to obtain. In the past, BSA and HSA were tested in humans for reducing osmotic pressure under a serious bleeding condition, but only HSA gave a positive result. SAs are quite large (~66 kDa), heart-shaped, and comprise three helical domains (I, II, and III). Each domain is constituted of two subdomains, A and B [107].

Electrochemical methods have been used to investigate the corrosion susceptibility of AZ91 magnesium surgical alloys in SBF consisting of BSA and acid SBF (pH 5). The addition of albumin to SBF has been shown to have a positive influence on the improvement of the open circuit potential (OCP). Moreover, the adsorbed BSA results in a lower cathodic current and higher corrosion resistance. Finally, a higher concentration of BSA is beneficial to mitigate the corrosion process of the magnesium alloy in the SBF [108].

#### 3.2.3. Gelatin

GEL is a natural biopolymer derived from collagen by partial acid or alkaline hydrolysis [109]. It is a natural imitation of the extracellular matrix (EMC) of human tissues and organs and is widely used in the field of tissue engineering due to its excellent biological origin, biocompatibility, biodegradability, non-immunogenicity, low antigenicity, the potential for chemical modification, adhesion, cell differentiation, and commercial availability at a relatively low cost [109,110,111]. However, GEL is a water-soluble protein, and crosslinking is usually needed to improve its mechanical properties and stability, making GEL scaffolds insoluble in biological environments [111]. An amorphous Mg67Zn28C5 alloy was coated with GEL by electrospinning to improve the cell surface interaction and biocompatibility of the amorphous Mg. The coated/uncoated alloys were immersed in a culture medium containing sodium bicarbonate and different concentrations of CO_2_ for 3 days. By varying CO_2_ in the immersion of the alloy, the pH, [Mg^2+^], and [Ca^2+^] were significantly affected via different mechanisms. Furthermore, the GEL electrospun-coated alloy exhibited a differential effect on the release of pH, Mg^2+^, and Zn^2+^ at different levels of CO_2_. The alloy extracts were not toxic to the coated/uncoated alloys, but the coated alloy showed binding of both cell types for 2 days [112].

Additionally, a coating was prepared on the surface of the micro-arc oxidation (MAO) film of the magnesium alloy WE42 by mixing different grades of cross-linked GEL with nanoparticles of poly (DL-lactic-co-glycolide) (PLGA). Potentiodynamic polarization was used to evaluate the corrosion behavior of the composite coating. The coating improved the corrosion resistance of the MAO film and WE42 Mg because the pores that emerged through the MAO layers were penetrated by the composite layer [113].

#### 3.2.4. Silk Fibroin

Silk fibroin (SF) is produced by *Bombyx mori* silkworms. SF is composed of two structural proteins, the fibroin heavy chain (~325 kDa) and the light chain (~25 kDa), linked together by a disulfide bond. These core fibers are encased in a sericin coat, a family of glue-like proteins that holds two fibroin fibers together and which represent 25 to 30% of the total weight of the silkworm cocoon [114,115]. Sericin has been associated with the immune response, but it can be easily removed by boiling alkaline solutions [116].

SF is composed of alternating hydrophobic and hydrophilic blocks and crystalline regions. The hydrophobic blocks consist of repeats of (Gly–Ala–Gly–Ala-Gly–Ser)_n_ that form nanocrystals rich in β-sheets [117]. The hydrophilic part of the core is non-repetitive, relatively elastic, and short [117,118]. Due to its amino acid sequence, SF provides opportunities for chemical modification. Amines, alcohols, phenols, carboxyl groups, and thiols have been explored as potentially reactive side groups for the chemical modification of SF [116].

The crystal structure of SF can take different forms because it has three types of polymorphisms [119], which are the glandular state (silk I); the spun silk state, which consists of the β-sheet secondary structure (silk II); and an air/water assembled interfacial silk (silk III, with a helical structure) [120]. The dominance of the β-sheet formation regimes within the fibroin structure confers high mechanical strength and toughness to protein-based materials [115].

Silkworm fibers are effective in biomedical applications due to their properties, including their biocompatibility, the ease with which they can be chemically modified, their slow rate of degradation in vivo [115], their low immunogenicity, and their limited bacterial adhesion [121]. Due to their versatility, various morphologies can be regenerated from dissolved fibroin fibers, such as sponges, hydrogels, films, microparticles, and microneedles [122]. Silk offers an attractive balance of modulus, breaking strength, and elongation, which contributes to its good toughness and ductility. Silk fibers are tougher than Kevlar, which is used as a benchmark in high-performance fiber technology [118]. Recent studies have shown that SF offers significant promise as a coating on Ti implants because the coated implant exhibited increased cell adhesion and mineralization without a significant immune response [123]. Thus, it is innovative and feasible to coat silk fibroin on the surface of magnesium alloys to achieve a coating of organic macromolecules and inorganic metal materials, thereby delaying the degradation rate of magnesium alloys [26].

A coating of SF and K_3_PO_4_ was deposited on Mg 1Ca alloy. SF performed as the skeleton coating, while the PO_4_^3−^ ions served as the corrosion inhibitor because of the lower solubility of the formed Mg_3_(PO_4_)_2_. The volume of hydrogen release and changes in pH value reflected the corrosion status of each group during 14-day immersion in Hank’s solution. The hydrogen release of the Silk-KP was an average of 1.19 mL/cm^2^, and the pH value was increased to 8.73 at 14 days. However, the bare Mg 1Ca alloy showed the quick and copious accumulation of hydrogen that evolved with time (5.75 mL/cm^2^ at 14 days), and the pH value was higher than 9 in the first three days and reached 10.13 at 14 days [124].

On the other hand, stents were coated with ethanol-treated SF to enrich the content of the β-sheet. The authors evaluated the release of a drug from the SF layer, the corrosion resistance of the Mg alloy, and the biocompatibility. The SF coating suppressed the local and deep corrosion of the Mg alloy stent; see Figure 3 [125].

The physical adhesion force between the natural organic film and the metal substrate when directly adhered to without any pretreatment will lead to the delamination of the film, because they cannot easily form strong chemical bonds. To solve this, a surface activation process was performed by short-wavelength vacuum ultraviolet (VUV/O_3_) activation and O_2_ plasma activation to increase the available functional groups on the Mg–Zn–Ca surface before coating with SF. The contact angle was measured, and the surface was found to be more hydrophilic when the contact angle was lower. The bare Mg–Zn–Ca surface possessed poor hydrophilicity with a contact angle value of 40°, because the surface was not clean enough. However, the contact angle values decreased sharply to 4.0° and 2.4° after O_2_ plasma activation and VUV/O_3_ activation, respectively. Therefore, the improved hydrophilicity due to the activation of VUV/O_3_ would provide more possibilities to obtain a strong adhesion force between coatings and substrates [56].

Similarly, SF was used as a natural organic polymer coating on an Mg–Zn–Ca alloy pretreated with 3-aminopropyltriethoxysilane (APTES). APTES pretreatment coats the surface of magnesium alloys with amino groups, which can bond with functional groups on silk fibroin to form a compact coating/substrate interface. The silk fibroin films reached a thickness of ~7 µm. During in vitro degradation and electrochemical measurements in simulated body fluid (SBF), the samples with the SF coating showed remarkably improved corrosion resistance and a slower degradation rate compared to nude samples. Furthermore, the excellent biocompatibility between SF and the substrate was confirmed [26].

In another study, the authors developed a silk fibroin/chitosan quaternary ammonium salt/heparin (SF/HACC/Hep) multilayer coating on the surface of AZ31B Mg alloy to enhance the corrosion resistance, biocompatibility, and antibacterial property for medical applications. The coating was degraded slowly in the corrosive medium and induced Ca-P formation. The multilayer coating also reduced the hemolysis rate and inhibited platelet adhesion, promoting endothelial cell adhesion and proliferation [126].

Elsewhere, authors modified a Mg–Zn–Ca Mg alloy coated with SF by hybrid activation plasma for orthopedic applications. The corrosion resistance was improved compared with bare Mg–Zn–Ca, not only because of the effective protection of the SF coating, but also thanks to its tight bonding on substrates [127].

Other researchers anodized ZK60 Mg alloy to form a micro-rough surface, followed by the layer-by-layer fabrication of a bioinorganic–bioorganic hybrid coating by surface mineralization with hydroxyapatite (HA) and SF and spin coating. The coating reduced the corrosion and degradation rate of ZK60, as well as enhancing the weight loss resistance. Hybrid coatings have the potential to improve the stability and longevity of Mg alloy implants [128].

#### 3.2.5. Fibrin and Fibrinogen

Fibrin is a natural biopolymer formed in the last step of the clotting cascade by the action of thrombin on fibrinogen. Fibrinogen is a large, complex, fibrous glycoprotein necessary to many biological processes, such as hemostasis, wound healing, inflammation, and angiogenesis, among others [129].

The degradation rates of iron and Mg with and without fibrin coatings were reported. The results showed that the inclusion of a fibrin coating did not change the nature of corrosion for the material, localized corrosion for iron, and more uniform corrosion in the case of Mg. The time to achieve complete corrosion was ~20 and 90 days for uncoated Mg and Fe wires; meanwhile, it was 40 and 200 days for fibrin-coated and Fe wires, respectively [130].

### 3.3. Organic Acids

#### 3.3.1. Phytic Acid

Phytic acid (PA) is a phosphate ester of inositol that has twelve hydroxyl and six phosphate carboxyl groups [58]. It is a non-toxic organic macromolecule and is widely present in nature (plants, animals, and soils), mainly as mixed salts of calcium (Ca), Mg, and potassium (K). The existence of this compound in seeds was first reported in 1903 and it is now accepted as being ubiquitous among plant seeds and grains, comprising 0.5–5% (*w*/*w*) [131].

Due to the structure of oxygen atoms, it has a powerful chelating capability with many metal ions, such as Zn^2+^, Fe^2+^, Fe^3+^, Ca^2+^, and Mg^2+^, to form stable metal–phytic complexes, which can be deposited on the surface of the metal substrate and thus improve the corrosion resistance [132]. PA is also known as an inexpensive and “green” reagent for the environment [133].

Some studies showed that the corrosion resistance properties of PA-coated Mg are comparable to or even better than a corresponding chromate conversion coating. The major determining parameters are the nature and extent of Mg ions available for chelation, the stability of the Mg (OH)_2_ layer, the concentration and conformation of PA, reaction time, bath pH and temperature, and pre-and post-treatments followed [59]. It was found that the highest extent of coating deposition was achieved at a PA concentration in the range of 5–15 g/L [59].

Another study obtained potentiodynamic polarization curves for phytic acid conversion coatings on AZ91D Mg alloys in a 3.5% NaCl solution. They showed that the open-circuit current density of the treated sample decreased approximately six orders more than that of the untreated sample. The anodic current density was lower for the phytic acid conversion coating than for the AZ91D magnesium alloy. This means that the coatings had much higher corrosion resistance than the substrate [134].

On the other hand, authors prepared a phytic acid/cerium composite coating (PA/Ce) with self-healing ability on AZ31B magnesium alloy by hydrothermal treatment. The self-healing coating could release Ce ions to the damaged zone and form a new transformation layer, which inhibited the further corrosion of the substrate. The composite coating showed self-healing behavior and could provide long-term corrosion protection for the substrate. It could significantly extend the service life of Mg alloy components [135].

#### 3.3.2. Stearic Acid

Stearic acid is a saturated fatty acid with long chains. It can be found in most animal and plant fats as a glycerol ester. The polar head group can bind with metal cations [12]. A coating of stearic acid on Mg plates grown by a hydrothermal treatment was studied. The corrosion resistance, studied by electrochemical methods, was much higher than in bare Mg. It was concluded that SA coating is a potential route by which to enhance the corrosion resistance of degradable Mg implants [136].

Hydroxyapatite has also been studied with stearic acid to produce a composite film on the surface of Mg alloys via a combination of phosphate conversion treatments. The contact angle test showed that the contact angle reached 154.5°. The superhydrophobic surface improved the corrosion resistance of the Mg [137].

An overview of the corrosion studies of natural coatings on Mg alloys is presented in Table 2. It is evident that few studies have performed in vivo tests; meanwhile, in vitro tests are found more often, and values of corrosion density (Icorr) and corrosion potential (Ecorr) are the most common indicators of corrosion behavior. However, these values differ in each piece of research.

### 3.4. Other Polymers 

In another study, seven natural polymers, namely dextran (Dex), carboxymethylcellulose (CMC), pectin, hydroxyethylcellulose (HEC), sodium alginate (ALG), chitosan (CS), and gum arabic (GA), were analyzed to determine the inhibition property of the AZ31 Mg alloy in a NaCl solution at 3.5 wt.% It was observed that, except for HEC and ALG, the natural polymers accelerated the corrosion of the AZ31 Mg alloy in the corrosive medium studied. This is because CMC, Dex, GA, CS, and PEC are chelating agents that bind with Mg^2+^ ions and may have formed in the electrolyte rather than on the Mg surface. The influence of the concentration on the performance of HEC and ALG was investigated in the electrochemical analysis. The concentrations of the polymers considered in this set of experiments were 0.5 g/L, 1 g/L, and 2 g/L. It was obtained that the two natural polymers behaved similarly, although HEC demonstrated better corrosion inhibition performance than ALG. The best inhibition performance was achieved with 1.0 g of the polymers because inhibitor molecules were available for adsorption, resulting in larger surface coverage. Meanwhile, with 0.5 g, the concentration of the polymer was not sufficient for significant adsorption, and with 2 g, the solution may have been saturated so that the distance between the individual molecules became too close, causing molecules’ aggregation. Again, the adsorbed molecules could interact with the un-adsorbed molecules and cause the detachment of the adsorbed species from the substrate surface. With this concentration, up to 64.13% and 58.27% inhibition efficiency is achievable for HEC and ALG, respectively [147].

Additionally, 151 individual chemical compounds were investigated to determine their inhibitory effects on AZ31, AZ91, AM50, WE43, ZE41, Elektron 21, and three grades of pure magnesium. The selection of potential inhibitors was based primarily on their ability to form soluble or precipitated complexes with Fe^2+^ or Fe^3+^. It was found that only 15 compounds exhibited inhibitory properties and, of these, more than 60% were compounds designated as toxic, carcinogenic, and harmful to the environment [148].

## 4. Conclusions

The main challenge for Mg and its alloys is to control the rate of degradation, which is caused when it encounters body fluids due to the influence of different factors, such as pH, alloying elements, concentration, and types of ions. This means that magnesium implant materials require sufficient mechanical strength and integrity during their time in the human body. Different studies have revealed that these factors can be controlled by modifying the surface and polymeric coatings.

The researched literature has shown that there are different types of biopolymers, among which the natural ones stand out due to their biomimetic nature, biocompatibility, and cell proliferation, because they provide barriers to limit direct contact between the surface of the Mg and the aqueous biological environment.

Medical applications of Mg and Mg alloy coatings are challenging, and it is difficult to achieve a uniform coating and an effective degradation rate. In many cases, a pretreatment on the Mg surface is required to achieve better adhesion and corrosion resistance. Different surface modification methods were reviewed to improve the corrosion resistance of magnesium or its alloys. Thus far, there are many techniques developed for the protection of the substrate; however, not all of them are used, since they are toxic or do not present biocompatibility. The most used technique in natural biopolymers is chemical conversion because it provides excellent adherence, attributed to the formation of chemical bonds with the substrate.

A candidate for biomedical implants must satisfy several requirements: it must offer mechanical support; degrade at a reasonable rate to be replaced by the new bone; favor cell adhesion, proliferation, and cellular differentiation; prevent infection, and exert a positive osteogenic effect. The latter makes the prevention of corrosion for Mg alloys with the release of osteoinductive factors and growth factors to speed up the healing process very attractive.

Indicators such as pH evolution, hydrogen release or Mg^+2^, changes in mechanical strength, and morphological changes are essential in measuring the corrosion behavior of the material. However, mechanical strength changes have not been widely investigated.

There are few investigations of natural coatings on Mg with in vivo assessments, which are necessary for future studies. In order to evaluate the potential of these materials in clinical applications, it is necessary to obtain more strong evidence of the optimal performance, qualitative research, and development, together with the collaboration of clinicians to obtain materials for specific uses.

Natural coatings are promising because they protect against corrosion, can be functionalized, improve osseointegration, and the body can metabolize the subproducts of the degradation as drug release. There are many efforts to diminish corrosion and the progress is promising, but more studies with other natural polymers, such as fibrin or stearic acid, which are more frequently studied with other metals, become necessary. Furthermore, more studies on drug delivery and the release of osteoinductive factors and growth factors are required.

## Figures and Tables

**Figure 1 polymers-14-05297-f001:**
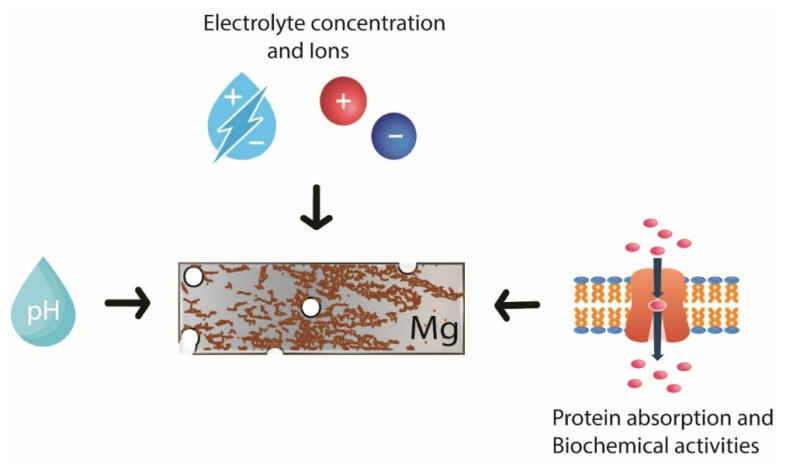
Influencing factors on Mg corrosion.

**Figure 2 polymers-14-05297-f002:**
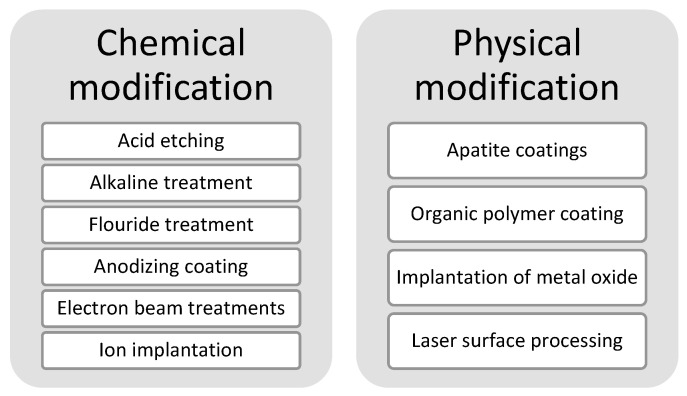
Methods of surface modification.

**Figure 3 polymers-14-05297-f003:**
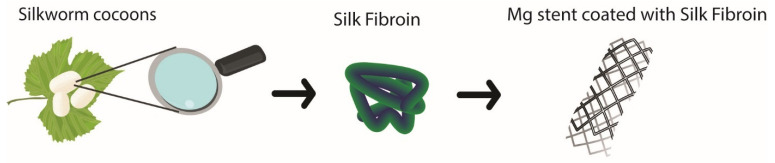
SF coated on Mg stent.

**Table 1 polymers-14-05297-t001:** Physical and mechanical properties of the main metals used for implants in comparison to natural bone.

Implant	Density (g/cm^3^)	Elastic Modulus (GPa)	Yield Strength (MPa)	Fracture Toughness (MPam^1/2^)	Ref
Natural bone	1.80–2.10	3–20	130–180	3–6	[16]
Mg	1.74–2.0	41–45	65–100	15–40	[17,18]
AZ31	1.78	41–45	185	N/A	[19]
Ti alloy	4.40–4.50	110–117	758–1117	55–115	[18]
Stainless steel	7.90	200	170–310	50–200	[19,20]
Co–Cr alloy	7–8	210–253	448–1606	N/A	[21]
PLA	1.25–1.29	2.2–3.3	N/A	N/A	[22]

No applicable (N/A).

**Table 2 polymers-14-05297-t002:** Natural coatings on Mg implants.

Coating	Material	Method	Treatment	Test		Comparison			Test		Applications	Ref
				Corrosion	Biological	Indicator	Coated	Uncoated	Cell line/Blood Type	Animal Model		
CS	AZ91E	Dipping	-	Potentiodynamic EIS Surface examination	-	Icorr (μA/cm^2^)	0.116	0.953	-	-	-	[68]
CS	Mg-1Ca	Dip-coating	Silanization	Immersion test in SBF	-	Hydrogen Release (mL/h cm^2^)	0.013	0.038	-	-	Biomedical	[69]
CS	Mg-4Li-1Ca	Dip-coating	MAO	Hydrogen evolution Potentiodynamic polarization EIS	-	Icorr (μA/cm^2^)	6.714	24.840	-	-	Bio-degradableOrthopedic implants	[70]
CS	Mg-6%Zn-10%Ca_3_(PO_4_)_2_	Smearing	-	In vitro corrosion	Cytocompatibility In vivo biodegradation	Hydrogen evolution (mL/cm^2^)	25	90	L-929 cells	Adult male Zelanian rabbits	Femoral implants	[138]
CS	Mg-10% tricalcium phosphate-6% Zinc	-	-	Immersion in SBF	Cytotoxicity	Hydrogen evolution (mL/cm^2^)	20	90	L-929 cells	-	Implantable devices	[39]
ALG/CS/Mechano-growth factor	ZEK	LBL self-assembly	Fluoride	Immersion test	Degradation in vivo	Loss of initial corrosion rate (%)	77.2	78.6	-	Rats	Femur bone repair	[74]
ALG/n-TiO_2_	Pure Mg	Electrophoretic deposition	-	Potentiodynamic EIS	-	Corrosion resistance (kΩ cm^2^)	1.6	0.4	-	-	Biomedical	[139]
Sodium alginate/Silane/Mg (OH)_2_	Pure Mg	Spin-coating	Alkali and silane	EIS Soaking test in SBF	-	Icorr (μA/cm^2^)	1.3	130.9	-	-	Temporary implants	[140]
Van-loaded/sodium alginate	AZ31D	Immersion	MAO	Potentiodynamic polarization EIS	Platelets adhesion Hemolysis ratio	Icorr (A/cm^2^)	1.68 × 10^−8^	6.25 × 10^−4^	Fresh human blood	-	Antibacterial Implantable material	[41]
PLA/CNs	AZ31	Dip-coating	-	Immersion test in SBF Dynamic polarization EIS	-	Icorr (μA/cm^2^)	0.19	28.8	-	-	Biomedical implants	[79]
CA	Mg-1Ca-0.2Mn-0.6Zr	Dipping	-	Potentiodynamic polarization	Cell viability/proliferation Morphology Histological	Icorr (μA/cm^2^)	4.88	497.96	MC3T3-E1 pre-osteoblasts	Male Albinos rats	Femur bone implants	[80]
Hydroxyapatite/CMC/Graphene	AZ31	Electrophoretic deposition	-	EIS Potentiodynamic polarization	-	Corrosion rate (mm/year)	0.0188	0.3418	-	-	Implant devices	[141]
HA/Ce (NO_3_)	Pure Mg	Electrodeposition	Hydrothermal	EIS	Cell viability In vivo	-	-	-	MC3T3-E1 osteoblasts	Tibia bones Sprague–Dawley rats	Implant devices	[84]
HA/CMC	Pure Mg	Hydrothermal immersion	Plasma electrolytic oxidation	Potentiodynamic polarization EIS Scratch and immersion test in SBF	Cytotoxicity In vivo	Icorr (A/cm^2^)	5.362 × 10^−7^	1.954 × 10^−5^	MC3T3-E1 osteoblasts	Femur male Sprague Dawley rats	Absorbable Mg screws	[85]
HA	AZ31	Dipping	Alkaline	Potential polarization Static immersion	Cytotoxicity Cell adhesion	Icorr (A/cm^2^)	5.71 × 10^−6^	8.48 × 10^−5^	MC3T3-E1 osteoblasts	-	Orthopedic implants	[142]
CHN/Hydroxyapatite	AZ91	Dipping	-	Potentiodynamic polarization	-	Icorr (μA/cm^2^)	10	24.5	-	-	Orthopedic implants	[87]
Hep/silane	AZ31	Dipping	-	Potentiodynamic polarization EIS	Platelet adhesion	Icorr (μA/cm^2^)	1.87	8.32	Fresh whole rat blood	-	Biodegradable metallic implants	[94]
SF/Hep/GREDVY	Mg–Zn–Y–Nd	Micro drop deposition	HF conversion	Long term immersion test EIS	Hemolysis rates Platelet adhesion In vitro cytotoxicity	Icorr (μA/cm^2^)	0.642	22.568	HUVECs ^1^ VSMCs ^2^	-	Vascular stents	[95]
Hep/Carboxymethyl chitosan	AZ31B	Dipping	Alkaline	Potentiodynamic polarization	Blood clotting analysis	Icorr (A/cm^2^)	4.29 × 10^−7^	1.18 × 10^−5^	Fresh human whole blood	-	Cardiovascular and orthopedic	[96]
Coll	Mg–Zr–Ca	Dip coating	-	Hydrogen production rate in SBF	Cell viability Bone embedding Histological and immunohistochemical analysis	-	-	-	MC3T3-E1 mouse osteoblasts	Male New Zealand white rabbits	Femur Bearing implant	[103]
Silane-TiO_2_/Coll	AZ31 ZE41	Dipping	HF	EIS	Cell viability Cell morphology	-	-	-	NHDF ^3^ HOb ^4^	-	Temporary metallic implants	[143]
Coll	Pure Mg MgYZrRE MgZnYZrRE	Soaking	HF	EIS pH change	In Vitro Endothelialization	pH	8.51 8.26 8.36	8.70 8.40 8.35	HCAEC ^5^	-	Cardiovascular stents	[42]
Albumin	AZ91	Immersion	-	Potentiodynamic polarization EIS immerse in SBF	-	Corrosion resistance (Ω cm^2^)	541.9	93.7	-	-	Bone implants	[108]
Albumin/APTES	Pure Mg	Silane linkers	-	Immersion in SBF	-	Hydrogen evolution (mL/cm^2^/d)	0.005	0.12	-	-	Non-permanent orthopedic implants	[144]
GEL	Mg–Zn–Ca	Layer by electrospinning	-	Immersion test	Cytotoxicity	-	-	-	MG63 osteosarcoma Mouse L929 fibrosarcoma	-	Orthopedic and cardiovascular prosthetic materials	[112]
GEL/n-PGLA	WE42	Dropwise	MAO	Potentiodynamic polarization EIS	-	Transfer resistance (kΩ cm^2^)	>100	3.5	-	-	Stents	[113]
CS/GEL/Bioactive glass	Mg-Si-Sr	Alternative current Electrophoretic deposition	-	Potentiodynamic polarization	-	Icorr (μA/cm^2^)	3.41	30.06	-	-	Screw, stents, wire, and orthopedic plates	[145]
GEL nanoparticles CS nanoparticles CAP nanoparticles	AZ91E	-	-	Potentiodynamic polarization EIS	In vivo corrosion	Icorr (μA/cm^2^)	1.461 8.686 8.484	15.837	-	White male rabbit	Drug delivery and Mg staples	[40]
SF/K_3_PO_4_	Mg-1Ca	Spin coating	Fluoride	In vitro corrosion EIS	Cytocompatibility Osteogenic activity	Hydrogen release (mL/cm^2^)	1.19	5.75	MC3T3-E1 mouse osteoblasts	-	Bone implants	[124]
SF/sirolimus	AZ31	Dip coating	HF EtOH	Immersion in E-MEM	Adhesion of endothelial cells and platelets	Mg^2+^ ion release rate (%)	7	30	HUVECs	-	bioresorbable stent implants	[125]
SF	MgZnCa	Dropwise	Vacuum ultraviolet ozone	Immersion in SBF Potentiodynamic polarization EIS	In vitro degradation Cell adhesion In vivo experiments	Icorr (μA/cm^2^)	1.41	12.36	BMSCs ^6^	Adult male New Zealand rabbits	Bone implants	[56]
SF	MgZnCa	Dropwise	APTES	Corrosion resistance in SBF EIS	Cell viability Cell adhesion Cell isolation	Ecorr (V)	-0.80	−1.56	BMSCs	-	Bone implants	[26]
SF/HACC/Hep	AZ31B	LBL self-assembly	Alkaline	Potentiodynamic Polarization EIS Immersion in SBF	Hemocompatibility Cytotoxicity	Icorr (A/cm^2^)	1.523 × 10^−6^	1.028 × 10^−5^	Human blood Endothelial cells	-	Cardiovascular and bone stents	[126]
SF	ZK60	LBL	-	Potentiodynamic polarization EIS In vitro degradation	-	Icorr (μA/cm^2^)	1.85	344.02	-	-	Implants	[128]
Fibrin	Pure Mg	Dropwise	-	Submersion in DMEM Optical examination	-	-	-	-	-	-	Bioabsorbable stents	[130]
PA	AZ91D	Chemical conversion	-	Potentiodynamic polarization Immersion test	-	Icorr (mA/cm^2^)	3.2	4.2	-	-	-	[146]
PA	AZ91D	Immersion	-	Potentiodynamic polarization EIS	-	-	-	-	-	-	-	[134]
PA/Ce	AZ31B	Dipping and drawing	Micro-arc oxidation	Potentiodynamic polarization EIS	-	Icorr (A/cm^2^)	1.24 × 10^−7^	7.90 × 10^−5^	-	-	Biomedical	[135]
Hydroxyapatite/stearic acid	ZK60	Phosphate conversion	-	Potentiodynamic polarization	-	Icorr (A/cm^2^)	2.48 × 10^−7^	2.48 × 10^−5^	-	-	Biomedical	[137]

^1^ Human umbilical vein endothelial cells (HUVECs). ^2^ Vascular smooth muscle cells (VSMCs). ^3^ Normal Human Dermal Fibroblasts (NHDF). ^4^ Human Osteoblasts (HOb). ^5^ Human Coronary Arteryc Endothelial Cells (HCAEC). ^6^ Bone marrow-derived mesenchymal stem/stromal cells (BMSC).

## Data Availability

Not applicable.

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
