# Peer review of "Natural Coatings and Surface Modifications on Magnesium Alloys for Biomedical Applications"

_polymers, 2022, doi:10.3390/polym14235297_

Round 1

Reviewer 1 Report

The researchers' need for such a review is not obvious. So, among the 116 references in the list of references, 60 are reviews on this topic, the last of which was published in 2021. I think that in the introduction it is necessary to emphasize the relevance of the presented review.

I also advise the authors to add the following articles to the list of references:

Bazhenov, V.; Lyskovich, A.; Li, A.; Bautin, V.; Komissarov, A.; Koltygin, A.; Bazlov, A.; Tokar, A.; Ten, D.; Mukhametshina, A.  Effect of Heat Treatment on the Mechanical and Corrosion Properties of Mg–Zn–Ga Biodegradable Mg Alloys.  Materials 2021, 14(24), 7847; https://doi.org/10.3390/ma14247847.

Kiselevsky, M.; Anisimova, N.; Polotsky, B.; Martynenko, N.; Lukyanova, E.; Sitdikova, S.; Dobatkin, S.; Estrin, Yu. Biodegradable magnesium alloys as promising materials for medical applications (review).  Modern technologies in medicine: 2019, 11, 146-157, DOI: 10.17691/stm2019.11.3.18.

Reviewer 2 Report

Natural coatings and surface modifications on magnesium alloys which makes this modification a promising candidate to demonstrate natural coating as a promising surface for biomedical applications

While the work is overall carried out well and the review support the conclusion, there are several issues that need attention and upon addressing those issues the paper can be accepted in Polymers

1-    Firstly, there are numerous typos (overtyping) throughout the manuscript, all requiring attention (the abstract has such errors).  There are several grammatical errors that needed to be corrected. I urge the authors to thoroughly go through the entire manuscript and check every line for spelling, grammar, or sentence construction-related errors as without these measures the account is unreadable. 

2-    In introduction: explain each word for first time in the begging and used its abbreviation after that

3-    Numerate all items to be correct (3.2. is repeated)

4-      Proper conclusion outcome of all items to be presented in the manuscript

Round 2

Reviewer 2 Report

Dear editor,

The authors replied to all comments

best regards

Sanaa